 SciPost Phys. Lect. Notes 17 (2020)

# Transport in one-dimensional integrable quantum systems

**Jesko Sirker**

Department of Physics and Astronomy, University of Manitoba, Winnipeg R3T 2N2, Canada

⋆ sirker@physics.umanitoba.ca

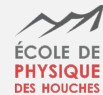 Part of the *Integrability in Atomic and Condensed Matter Physics Collection*
*Session 111 of the Les Houches School, August 2018*
*published in the Les Houches Lecture Notes Series*

## Abstract

These notes are based on a series of three lectures given at the Les Houches summer school on 'Integrability in Atomic and Condensed Matter Physics' in August 2018. They provide an introduction into the unusual transport properties of integrable models in the linear response regime focussing, in particular, on the spin-1/2 XXZ spin chain.

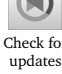 Check for updates

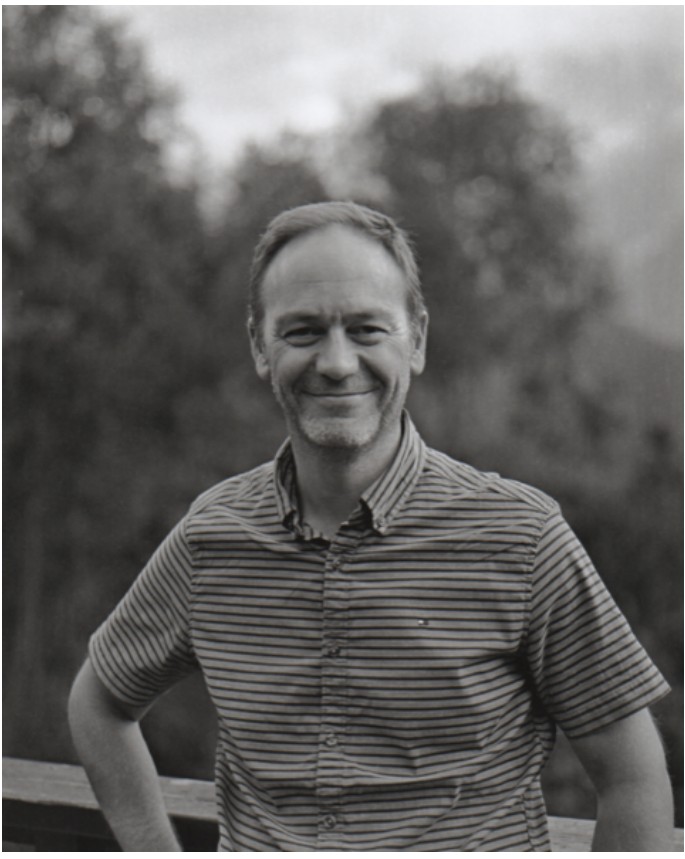

# 1 Outline

In these lecture notes I will discuss transport in one-dimensional quantum systems at finite temperatures in the linear response regime. After a general introduction, a particular focus will be on the unusual transport properties of integrable systems. Note that these notes are not meant to be an exhaustive review of the research field. They are based on the content of three lectures given at the Les Houches summer school on 'Integrability in Atomic and Condensed Matter Physics' in August 2018 and are therefore necessarily limited in scope. These lectures on transport build on material which was presented at the summer school in earlier lectures. Foundations of the coordinate, algebraic, and thermodynamic Bethe ansatz, in particular, are assumed to be known already. Furthermore, I also note that a different approach to transport—generalized hydrodynamics—has been discussed in a separate series of lectures and will not be covered here.

In order to be concrete, I will mostly concentrate on a particular integrable lattice model, the XXZ spin chain

$$H = \sum_\ell \left[ J \left( S_\ell^x S_{\ell+1}^x + S_\ell^y S_{\ell+1}^y + \Delta S_\ell^z S_{\ell+1}^z \right) - h S_\ell^z \right]. \tag{1.1}$$

Here $J$ is the exchange constant, $\Delta$ the exchange anisotropy, and $h$ an external magnetic field. $S^\alpha$ are spin-1/2 operators fulfilling the commutation relations $[S^\alpha, S^\beta] = i\varepsilon_{\alpha\beta\gamma}S^\gamma$. In the following, we will often parametrize the anisotropy as $\Delta = \cos\gamma$. It is also often useful to think about this model as a chain of interacting spinless fermions

$$H = \sum_\ell \left\{ J \left[ -\frac{1}{2}(c_\ell^\dagger c_{\ell+1} + h.c.) + \Delta \left( n_\ell - \frac{1}{2} \right) \left( n_{\ell+1} - \frac{1}{2} \right) \right] - h \left( n_\ell - \frac{1}{2} \right) \right\}, \tag{1.2}$$

with $n_\ell = c_\ell^\dagger c_\ell$ where $c_\ell$ is a fermionic annihilation operator at site $\ell$. This alternative representation is obtained by using the Jordan-Wigner transformation

$$S_\ell^z \to n_\ell - \frac{1}{2}, \quad S_\ell^+ \to (-1)^\ell c_\ell^\dagger e^{i\pi\phi_\ell}, \quad S_\ell^- \to (-1)^\ell c_\ell e^{-i\pi\phi_\ell}, \tag{1.3}$$

with the ladder operators $S_\ell^\pm = S_\ell^x \pm iS_\ell^y$ and the Jordan-Wigner string $\phi_\ell = \sum_{j=1}^{\ell-1} n_j$. Note that the Jordan-Wigner string does not show up explicitly in the Hamiltonian (1.2) because the hopping is limited to nearest-neighbour sites in the lattice.

In general, transport is a *non-equilibrium problem*: Spin transport, for example, requires a magnetic field gradient while heat transport is driven by a temperature gradient. In the following I will, however, exclusively analyze the linear response regime using *Kubo formulas*. In this regime, transport coefficients can be obtained from dynamical correlation functions *calculated at equilibrium*. The plan for the three lectures is then as follows: In the first lecture, I will briefly recapitulate how currents and transport coefficients can be defined. Furthermore, I will derive the Mazur inequality and explain why integrability can lead to ballistic transport even at finite temperatures. In the second lecture, the Kubo formulas for the conductivities will be discussed. I will show, in particular, how the Drude weight and the diffusion constant can be obtained from real-time equilibrium current-current correlation functions. Explicit results for the thermal Drude weight of the XXZ chain will be derived. To obtain a broader physical understanding of the interplay of ballistic and diffusive transport channels, I will describe the XXZ chain at low energies using bosonization in the third lecture. Finally, I will use the field theoretical description to calculate the spin conductivity, obtain a concrete formula for the spin diffusion constant, and discuss the general picture emerging from these calculations as well as their limitations.

## 2 Transport coefficients and linear response

One way to derive the spin and thermal current operators for the XXZ chain is based on a discrete version of the continuity equation where the time derivative is calculated using the equation of motion. For the total spin current $\mathcal{J}^s = \sum_\ell j_\ell^s$ we have, in particular,

$$\partial_t S_\ell^z = -i[S_\ell^z, H] = -(j_\ell^s - j_{\ell-1}^s) \tag{2.1}$$

leading to a current density

$$j_\ell^s = J(S_\ell^x S_{\ell+1}^y - S_\ell^y S_{\ell+1}^x) = \frac{iJ}{2}(S_\ell^+ S_{l+1}^- - S_\ell^- S_{l+1}^+). \tag{2.2}$$

Using the Jordan-Wigner transformation (1.3) we see that in terms of spinless fermions this corresponds to a particle current, i.e., the difference between particles moving to the left and to the right.

Similarly, we can derive the thermal current operator $\mathcal{J}^{\text{th}} = \sum_\ell j_\ell^{\text{th}}$ by the continuity equation

$$\partial_t h_{\ell,\ell+1} = -i[h_{\ell,\ell+1}, H] = -(j_\ell^{\text{th}} - j_{\ell-1}^{\text{th}}), \tag{2.3}$$

where $H = H^0 - h\sum_\ell S_\ell^z = \sum_\ell h_{\ell,\ell+1} = \sum_\ell(h_{\ell,\ell+1}^0 - hS_\ell^z)$. The thermal current thus splits into two parts, $J^{\text{th}} = J^E - hJ^s$, where $J^s$ is the spin current (2.2) and $J^E$ the energy current obtained from the continuity equation (2.3) for the case of zero magnetic field. In other words, at finite magnetic fields there is a contribution to the thermal current due to particle transport. Calculating the commutator in (2.3) for $h = 0$, leads to an energy current density $j_\ell^E$ acting on three neighbouring sites which can be written in compact form as

$$j_\ell^E = J^2 \sum_\ell \mathbf{S}_\ell \cdot (\mathbf{S}_{\ell-1}' \times \mathbf{S}_{\ell+1}'), \quad \mathbf{S}_\ell' = (S_\ell^x, S_\ell^y, \Delta S_\ell^z). \tag{2.4}$$

Alternatively, the spin current can also be derived by putting a flux $\Phi$ through an XXZ ring in the fermionic formulation (1.2). The flux then couples via the Peierls substitution

$c^\dagger_\ell c_{\ell+1} \to c^\dagger_\ell c_{\ell+1} e^{-iA_{\ell,\ell+1}}$. Here $A_{\ell,\ell+1}$ is the vector potential along the bond with $\sum_\ell A_{\ell,\ell+1} = \Phi$. The current operator is then given by $j^s_\ell = -\frac{\partial H}{\partial A_{\ell,\ell+1}}\big|_{A\to 0}$. Furthermore, the diamagnetic term can be obtained as $\frac{\partial^2 H}{\partial A^2}\big|_{A\to 0} = H_{\text{kin}}$ where $H_{\text{kin}}$ is the hopping part of the Hamiltonian (1.2).

The transport coefficients relate the currents to the gradients in temperature and magnetic field

$$\begin{pmatrix} \mathcal{J}^{\text{th}} \\ \mathcal{J}^s \end{pmatrix} = \begin{pmatrix} \kappa_{\text{th}} & C^{\text{th}}_s \\ C^s_{\text{th}} & \sigma_s \end{pmatrix} \begin{pmatrix} -\nabla T \\ \nabla h \end{pmatrix}, \tag{2.5}$$

with $\kappa_{\text{th}}$ being the thermal conductivity and $\sigma_s$ the spin conductivity. The coefficients $C^{\text{th}}_s$ and $C^s_{\text{th}}$ describe the creation of a thermal current due to a magnetic field gradient and of a spin current due to a thermal gradient, respectively. The latter is the spin Seebeck effect which has been studied in much detail for ferromagnets in the field of spintronics. From the Onsager relation [1] it follows that $C^{\text{th}}_s = T C^s_{\text{th}}$.

The, in general, complex and frequency dependent transport coefficients are decomposed as, for example,

$$\sigma'_s(k=0,\omega) = 2\pi D_s \delta(\omega) + \sigma^{\text{reg}}_s(\omega), \tag{2.6}$$

where $\sigma'_s(k,\omega)$ denotes the real part of the spin conductivity at momentum $k$ and frequency $\omega$. $D_s$ is the *spin Drude weight*, and $\sigma^{\text{reg}}_s(\omega)$ the regular part of the conductivity. We can write down a similar decomposition for the thermal conductivity $\kappa_{\text{th}}(\omega)$. A non-zero Drude weight signals *ballistic transport*, i.e. a diverging dc conductivity. Physically, this means that the current does not completely relax. In a lattice system without impurities, we expect this to happen at zero temperature where scattering processes such as spin-spin or spin-phonon are frozen out. At finite temperatures, on the other hand, we expect that in a generic clean system the dc conductivity becomes finite. Scattering processes are expected to lead to a temperature-dependent broadening of the delta peak. This can only be avoided if a part of the current is fully protected from relaxing by some conservation law.

This is the point where integrability comes into play. What makes integrable models special, is that they have an *infinite set of local conserved charges* $\mathcal{Q}_j$. Here we mean local in the strict sense that

$$\mathcal{Q}_j = \sum_\ell q^j_\ell, \tag{2.7}$$

where $q^j$ is a local charge density acting on $j$ neighbouring sites. For the XXZ chain, in particular, we can derive these charges by defining a family of commuting transfer matrices $[T(\theta), T(\theta')] = 0$ with spectral parameter $\theta$. The local conserved charges are then obtained by

$$\mathcal{Q}_{j+1} = \frac{d^j}{d\theta^j} \ln T(\theta)\big|_{\theta=1}, \quad j \geq 1. \tag{2.8}$$

We refer to reference [2] for details. For now it is just important to note that $\mathcal{Q}_2 \propto H$ and $\mathcal{Q}_3 \propto \mathcal{J}^E$. Thus the energy current is itself a conserved quantity implying an infinite energy conductivity at any temperature. Energy transport in the XXZ chain is purely ballistic.

Spin transport, on the other hand, is a much more complicated phenomenon. For zero magnetic field, the spin inversion operation $\mathcal{C}^{-1}\sigma^z_\ell \mathcal{C} = -\sigma^z_\ell$, $\mathcal{C}^{-1}\sigma^\pm_\ell \mathcal{C} = \sigma^\mp_\ell$ is a symmetry of the Hamiltonian. It is also easy to see that $\mathcal{C}^{-1}\mathcal{J}^E\mathcal{C} = \mathcal{J}^E$. For $h=0$ one can, furthermore, show that the transfer matrix $T(\theta)$ in the usual spin $s=1/2$ representation of auxiliary space is itself even under spin inversion for all spectral parameters $\theta \neq 0, \infty$, i.e. $\mathcal{C}^{-1}T(\theta)\mathcal{C} = T(\theta)$. Therefore *all* the local conserved charges $\mathcal{Q}_j$ defined in Eq. (2.8) are even under spin inversion. The spin current operator, on the other hand, is odd, $\mathcal{C}^{-1}\mathcal{J}^s\mathcal{C} = -\mathcal{J}^s$. It follows that for zero magnetic field $\langle \mathcal{J}^s \mathcal{Q}_j \rangle \equiv 0$, $\forall j$ where $\langle \cdots \rangle$ denotes the thermal average. Therefore the charges $\mathcal{Q}_j$ do not protect the spin current $\mathcal{J}^s$ from decaying. This leads to the interesting question whether or not spin transport in the XXZ chain at zero magnetic field is ballistic or diffusive.

We will see in the following that the answer depends on the anisotropy $\Delta$. For $\Delta = \cos(\pi/m)$ and $m$ integer, in particular, the two transport channels coexist. While we concentrate on these specific anisotropies in the following, we note that the results can be generalized to all commensurate anisotropies $\gamma = n\pi/m$ with $n, m$ coprime. On the other hand, it has been argued that ballistic transport possibly coexists with superdiffusion at incommensurate anisotropies while transport is entirely superdiffusive at the isotropic point, $\Delta = 1$ [3]. Also note that the arguments above do not apply to the case $h \neq 0$ where spin inversion $\mathcal{C}$ is no longer a symmetry of the Hamiltonian (1.1). In the latter case it is straightforward to show that a part of the spin current is protected by the conservation laws (2.8) and cannot decay [4].

## 2.1 Mazur inequality

To understand more precisely the connection between the Drude weight and the conserved charges of the considered system, we follow the approach of Mazur [5] and Suzuki [6]. We do so starting with a finite system. This raises some subtle questions with regard to the order of taking the limits of system size and time to infinity. We will get back to this point at the end of this section.

Let us start by considering the time average of a current-current correlation in spectral representation

$$
\lim_{\Lambda \to \infty} \frac{1}{\Lambda} \int_0^\Lambda dt \, \langle \mathcal{J}(t)\mathcal{J}(0) \rangle = \sum_{n,m} \frac{\mathrm{e}^{-\beta E_n}}{Z} \langle n|\mathcal{J}|m \rangle \langle m|\mathcal{J}|n \rangle \lim_{\Lambda \to \infty} \frac{1}{\Lambda} \int_0^\Lambda dt \, \mathrm{e}^{it(E_n - E_m)}
$$
$$
= \sum_{n,m}^{E_n = E_m} \frac{\mathrm{e}^{-\beta E_n}}{Z} |\langle n|\mathcal{J}|m \rangle|^2 , \tag{2.9}
$$

where $Z = \mathrm{tr}\{\mathrm{e}^{-\beta H}\}$ is the partition function. Here we have used that taking the limit yields

$$
\lim_{\Lambda \to \infty} \frac{\mathrm{e}^{i\Lambda(E_n - E_m)} - 1}{i\Lambda(E_n - E_m)} = \begin{cases} 0, & E_n \neq E_m \\ 1, & E_n = E_m \end{cases} . \tag{2.10}
$$

Without loss of generality, we can assume that we have a complete set of Hermitian conserved charges $Q_k$, $[H, Q_k] = 0$, which are orthogonal $\langle Q_k Q_l \rangle = \langle Q_k^2 \rangle \delta_{kl}$. We can then split the current operator into a part which is diagonal in the energy eigenbasis and a part which is off-diagonal. The diagonal part can then be expanded in $Q_k$:

$$
\mathcal{J} = \sum_k a_k Q_k + \mathcal{J}', \quad \text{with } \langle n|\mathcal{J}'|m \rangle = 0 \text{ if } E_n = E_m
$$
$$
\Rightarrow \langle Q_l \mathcal{J} \rangle = \sum_k a_k \underbrace{\langle Q_l Q_k \rangle}_{\langle Q_l^2 \rangle \delta_{k,l}} + \underbrace{\langle Q_l \mathcal{J}' \rangle}_{=0} \quad \Rightarrow a_l = \frac{\langle Q_l \mathcal{J} \rangle}{\langle Q_l^2 \rangle}. \tag{2.11}
$$

Keeping in mind that $\langle n|\mathcal{J}'|m \rangle = 0$ if $E_n = E_m$, we can therefore write the time average as

$$
(2.9) = \sum_{n,m}^{E_n = E_m} \frac{\mathrm{e}^{-\beta E_n}}{Z} \sum_{k,l} \frac{\langle \mathcal{J} Q_k \rangle \langle \mathcal{J} Q_l \rangle}{\langle Q_k^2 \rangle \langle Q_l^2 \rangle} \langle n|Q_k|m \rangle \langle m|Q_l|n \rangle. \tag{2.12}
$$

The $Q_k$ are diagonal and therefore

$$
\sum_{n,m}^{E_n = E_m} \frac{\mathrm{e}^{-\beta E_n}}{Z} \langle n|Q_k|m \rangle \langle m|Q_l|n \rangle = \sum_{n,m} \frac{\mathrm{e}^{-\beta E_n}}{Z} \langle n|Q_k|m \rangle \langle m|Q_l|n \rangle \tag{2.13}
$$
$$
= \sum_n \frac{\mathrm{e}^{-\beta E_n}}{Z} \langle n|Q_k Q_l|n \rangle = \langle Q_k Q_l \rangle = \delta_{kl} \langle Q_k^2 \rangle .
$$

This leads us to the final result

$$\lim_{\Lambda \to \infty} \frac{1}{\Lambda} \int_0^\Lambda dt \, \langle \mathcal{J}(t)\mathcal{J}(0)\rangle = \sum_k \frac{\langle \mathcal{J}Q_k\rangle^2}{\langle Q_k^2\rangle} \,. \tag{2.14}$$

If we find any conserved charge with $\langle \mathcal{J}Q_k\rangle \neq 0$ then (2.14) provides a lower bound for the time-averaged current-current correlation function in a finite system because the r.h.s. of Eq. (2.14) is strictly positive. The relation is then called the *Mazur inequality* and the obtained bound the *Mazur bound*.

In the thermodynamic limit, $N \to \infty$, we expect the current-current correlation function to equilibrate. If this is the case, then the time average becomes dominated by the constant equilibrium value, thus

$$\lim_{t \to \infty} \lim_{N \to \infty} \frac{1}{2NT} \langle \mathcal{J}(t)\mathcal{J}(0)\rangle = \lim_{N \to \infty} \frac{1}{2NT} \sum_k \frac{\langle \mathcal{J}Q_k\rangle^2}{\langle Q_k^2\rangle} \,. \tag{2.15}$$

In writing Eq. (2.15) we take for granted that the Mazur equality remains valid in the thermodynamic limit, i.e., that we can take the limit $N \to \infty$ first before taking $t \to \infty$ as is required in thermodynamics. Physically this is fairly obvious since the current density-density correlator $\langle j_\ell(t)j_0(0)\rangle$ is only non-zero (up to exponentially small tails) within the light cone set by the Lieb-Robinson bounds. I.e., for any time $t$ it is sufficient to consider a finite system of size $N \gg v_{LR}t$ where $v_{LR}$ is the Lieb-Robinson velocity. This point is discussed in more detail in Ref. [7]. We will see later that Eq. (2.15) is proportional to the Drude weight $D(T)$.

If $\mathcal{J}$ is a local operator—this is the case for the XXZ chain considered here— then $\langle \mathcal{J}Q_k\rangle^2 \sim N^2$. Therefore only those conserved charges contribute to the Mazur bound in the thermodynamic limit for which

$$\langle Q_k^2\rangle \sim N \,. \tag{2.16}$$

Operators who fulfill the strict locality condition, Eq. (2.7), also fulfill the condition (2.16). Additional conserved charges, however, can exist which are not of the form (2.7) but do fulfill Eq. (2.16). These charges are sometimes called *quasi-local* and play an important role in understanding the spin transport properties of the XXZ chain. In addition to conserved charges which are local in the sense of Eq. (2.16), every quantum mechanical system also has an infinite number of non-local conserved charges. An example are the projectors $P_n = |n\rangle\langle n|$ onto the extended eigenstates $|n\rangle$ of the system. Such charges, however, do not affect the transport properties of the system.

## 2.2 Kubo formula

Next, we want to discuss how to calculate the spin conductivity $\sigma_s(\omega)$ in linear response and how to relate Eq. (2.15) to the Drude weight. The Kubo formula is obtained straightforwardly in linear response theory and is given by

$$\sigma_s(\omega) = \frac{i}{\omega} \left[ \frac{\langle H_{\mathrm{kin}}\rangle}{N} - \frac{i}{N} \int_0^\infty dt \, e^{i\omega t} \langle [\mathcal{J}^s(t), \mathcal{J}^s(0)]\rangle \right]. \tag{2.17}$$

The first term is the diamagnetic contribution while the second term is the retarded current-current correlation function. For a derivation see, for example, the textbook by Mahan [1]. Using again a spectral representation, we can perform the integral over time and obtain

$$\sigma_s(\omega) = \frac{i}{\omega N} \left[ \langle H_{\mathrm{kin}}\rangle + \sum_{n,m} \frac{(p_n - p_m)|\langle n|\mathcal{J}^s|m\rangle|^2}{\omega - (E_m - E_n) + i\delta} \right], \tag{2.18}$$

with $p_n = \exp(-\beta E_n)/Z$ and $\beta = 1/T$. We now use the relation

$$\frac{1}{\omega}\frac{1}{\omega+E} = \frac{1}{E}\left(\frac{1}{\omega} - \frac{1}{\omega+E}\right) \tag{2.19}$$

to split Eq. (2.18) into two parts

$$\sigma_s(\omega) = \frac{i}{\omega N}\left[\langle H_{\text{kin}}\rangle + \sum_{n,m}\frac{(p_n-p_m)}{E_n-E_m}|\langle n|\mathcal{J}^s|m\rangle|^2\right] - \frac{i}{N}\sum_{n,m}\frac{(p_n-p_m)}{E_n-E_m}\frac{|\langle n|\mathcal{J}^s|m\rangle|^2}{\omega-(E_m-E_n)}. \tag{2.20}$$

The term in the square brackets is the charge or Meissner stiffness $\Gamma_s$. It can be obtained from the free energy $f(\Phi)$ of an XXZ ring with a flux $\Phi$ through the ring by $\Gamma_s = \frac{\partial^2 f}{\partial \Phi^2}\big|_{\Phi=0}$. The charge stiffness is proportional to the superfluid density $n_s(T)$ which is zero in the thermodynamic limit for a strictly one-dimensional system, see Refs. [8, 9] for details.

We now take the real part of the last term in Eq. (2.20) using the relation

$$\frac{1}{\omega-E} = P\frac{1}{\omega-E} - i\pi\delta(\omega-E) \tag{2.21}$$

to obtain

$$\sigma_s'(\omega) = -\frac{\pi}{N}\sum_{n,m}\frac{p_n-p_m}{E_n-E_m}|\langle n|\mathcal{J}^2|m\rangle|^2\delta(\omega-(E_m-E_n)) \tag{2.22}$$

$$= \frac{\beta\pi}{N}\sum_{E_n=E_m}p_n|\langle n|\mathcal{J}^2|m\rangle|^2\delta(\omega) + \frac{\pi}{N}\sum_{E_n\neq E_m}\frac{p_n-p_m}{E_m-E_n}|\langle n|\mathcal{J}^2|m\rangle|^2\delta(\omega-(E_m-E_n)).$$

Comparing with Eq. (2.6) we see that the first term in the second line is proportional to the Drude weight while the second term describes the regular part.

Using a spectral representation it is also straightforward to show that Eq. (2.22) can be rewritten as a time-dependent current-current correlation function

$$\sigma_s'(\omega) = \frac{1-e^{-\beta\omega}}{2\omega N}\int_{-\infty}^{\infty}e^{i\omega t}\langle\mathcal{J}^s(t)\mathcal{J}^s(0)\rangle. \tag{2.23}$$

This relation is known as the fluctuation-dissipation theorem because for generic, non-integrable models it connects the current-current fluctuations to the dissipative part of the conductivity.

For an integrable system, we can further split the correlation function into a ballistic part which persists at infinite times and a regular part which decays in time

$$C(t) = \lim_{N\to\infty}\langle\mathcal{J}^s(t)\mathcal{J}^s(0)\rangle/N = \underbrace{\lim_{t\to\infty}\lim_{N\to\infty}\langle\mathcal{J}^s(t)\mathcal{J}^s(0)\rangle/N}_{(\mathcal{J}^s\mathcal{J}^s)_\infty} + C_s^{\text{reg}}(t). \tag{2.24}$$

Here $C_s^{\text{reg}}(t)$ is a function which vanishes for $t\to\infty$ and gives a non-singular contribution to the conductivity $\sigma_s'(\omega)$. Plugging (2.24) into (2.23) yields

$$\begin{aligned}\sigma_s'(\omega) &= \frac{1-e^{-\beta\omega}}{2\omega}\int_{-\infty}^{\infty}dt\,e^{i\omega t}\left[(\mathcal{J}^s\mathcal{J}^s)_\infty + C_s^{\text{reg}}(t)\right]\\ &= 2\pi\frac{(\mathcal{J}\mathcal{J})_\infty}{2T}\delta(\omega) + \frac{1-e^{-\beta\omega}}{2\omega}C_s^{\text{reg}}(\omega).\end{aligned} \tag{2.25}$$

Comparing with the definition of the Drude weight and the regular part of the conductivity (2.6) we find the important relation

$$D_s = \frac{(\mathcal{J}^s\mathcal{J}^s)_\infty}{2T} = \lim_{t\to\infty}\lim_{N\to\infty}\frac{1}{2NT}\langle\mathcal{J}^s(t)\mathcal{J}^s(0)\rangle. \tag{2.26}$$

I.e., we have now shown that the expression in (2.15) is indeed the Drude weight and that this quantity is directly related to the part of the current which does not decay. Furthermore,

$$\sigma_s^{\mathrm{reg}}(\omega \to 0) = \beta \int_0^\infty dt\, C_s^{\mathrm{reg}}(t) = \chi_s(\beta)\mathcal{D}_s, \qquad (2.27)$$

where we have used the Einstein relation in the second step to introduce the *diffusion constant* $\mathcal{D}_s$ and the static spin susceptibility $\chi_s$. In addition to the Drude weight which is related via Eq. (2.26) to the part of the current which is protected by local conservation laws and does not decay in time, there is thus a diffusive part given by the decaying part of the current with diffusion constant

$$\mathcal{D}_s = \frac{\beta}{\chi(\beta)} \int_0^\infty dt\, [C(t) - 2TD_s]. \qquad (2.28)$$

We note that we have assumed here that the integral in Eq. (2.28) is convergent. If this is not the case, then the additional channel is superdiffusive. As indicated earlier, this possibly happens at incommensurate anisotropies but will not be discussed here any further. We can now combine (2.26) with the Mazur formula (2.15) to obtain a bound or the exact Drude weight by considering overlaps of the conserved charges with the current operator. The advantage of this approach is that it maps a dynamic onto a static problem. This approach has been used in Refs. [2, 10–12]. It provides a strict lower bound—possibly even exhaustive—for rational $\gamma/\pi$ and thus proof that ballistic transport for these anisotropies indeed persists at finite temperatures.

Similar results can also be obtained for the thermal conductivity. A subtle point is the proper definition of the currents and forces which cause these currents to flow, see Ref. [1]. One possible choice is $\mathcal{J}^s = \frac{M^{11}}{T}\nabla h$ and $\mathcal{J}^E = M^{22}\nabla\left(\frac{1}{T}\right)$. Comparing with (2.5) we see that there is an additional factor of $1/T$ in the definition of the thermal conductivity $\kappa_{\mathrm{th}}$. For the thermal Drude weight at zero field one finds, in particular,

$$D_{\mathrm{th}} = \lim_{t \to \infty} \lim_{N \to \infty} \frac{1}{2NT^2} \lim_{t \to \infty} \langle \mathcal{J}^E(t)\mathcal{J}^E(0) \rangle = \lim_{N \to \infty} \frac{\langle (\mathcal{J}^E)^2 \rangle}{2NT^2}, \qquad (2.29)$$

where we have used in the last step that $[\mathcal{J}^E, H] = 0$ for the XXZ chain.

## 3 Thermal Drude weight

The thermal Drude weight is particularly easy to calculate because it is given by the static expectation value of a conserved charge, see (2.29). In the following we briefly sketch how to obtain $D_{\mathrm{th}}$ using the standard thermodynamic Bethe ansatz (TBA) formalism for anisotropies $\Delta = \cos\gamma$ with $\gamma = \pi/m$. We note that the first derivation of the thermal Drude weight was carried out by Klümper and Sakai [13] using the quantum transfer matrix formalism. The latter approach has the advantage that the string hypothesis is not needed and results for arbitrary $\Delta$ are obtained.

We consider only the case $h = 0$. First, we define a generalized partition function and generalized free energy

$$Z = \mathrm{tr}\exp(-\beta H + \lambda J^E), \quad f(\beta, \lambda) = -\frac{T}{N}\ln Z. \qquad (3.1)$$

In TBA we can write this free energy density as

$$f(\beta, \lambda) = -\frac{T}{2\pi} \sum_{\ell=1}^m \int d\theta\, \varepsilon_\ell(\theta)\sigma_\ell \ln[1 + \eta_\ell^{-1}(\theta)]. \qquad (3.2)$$

Here $\varepsilon_\ell$ are the bare eigenenergies. The variables $\sigma_\ell = \text{sign}(g_\ell)$ are the signs of auxiliary rational numbers associated to string solutions as defined in [14]. For the case of anisotropy $\gamma = \pi/m$ the $g_\ell$ have a particularly simple relation to string length $n_\ell$

$$g_\ell = m - n_\ell, n_\ell = \ell \text{ for } \ell = 1, \ldots, m-1 \text{ and } g_m = -1, n_m = 1. \tag{3.3}$$

The functions $\eta_\ell = \rho_\ell^h/\rho_\ell$ are defined by the ratio of hole density $\rho_\ell^h$ and particle density $\rho_\ell$ of the $\ell$-th particle (string) and fulfill the coupled TBA equations

$$\ln \eta_\ell(\theta) = \beta \varepsilon_\ell + \lambda j_\ell^E + \sum_\kappa \int d\mu K_{\ell\kappa}(\theta - \mu) \sigma_\kappa \ln(1 + \eta_\kappa^{-1}(\mu)),$$
$$\equiv \beta \varepsilon_\ell + \lambda j_\ell^E + \left[ K * \sigma \ln(1 + \eta^{-1}) \right]_\ell, \tag{3.4}$$

with an integration kernel $K$, and '*' denoting a convolution and sum over Bethe strings. Here $j_\ell^E = \partial_\theta \varepsilon_\ell = \partial_\theta^2 p_\ell = p_\ell''$ where $p(\theta)$ is the momentum. To express the results in a more compact form, it is useful to define the following dressed quantities

$$\widetilde{\varepsilon}_\ell = \varepsilon_\ell - [K * \sigma \vartheta \widetilde{\varepsilon}]_\ell, \quad \widetilde{j}_\ell^E = j_\ell^E - [K * \sigma \vartheta \widetilde{j}^E]_\ell, \tag{3.5}$$

where we have defined the Fermi factor $\vartheta_\ell = 1/(1 + \eta_\ell) = \rho_\ell/(\rho_\ell + \rho_\ell^h)$. It is also useful to realize the following simple relation of the dressed quantities to the logarithmic derivatives of the $\eta$-functions

$$\partial_\beta \log \eta_\ell(\theta) = \widetilde{\varepsilon}_\ell(\theta), \quad \partial_\lambda \log \eta_\ell(\theta) = \widetilde{j}_\ell^E(\theta). \tag{3.6}$$

It is now straightforward to obtain the expectation value needed to calculate the thermal Drude weight

$$\langle (J^E)^2 \rangle / N = -\frac{1}{T} \partial_\lambda^2 f(\beta, \lambda)|_{\lambda=0} = \frac{1}{2\pi} \sum_\ell \int d\theta \, \varepsilon_\ell \sigma_\ell \partial_\lambda^2 \ln(1 + \eta_\ell^{-1})$$
$$= \frac{1}{2\pi} \sum_\ell \int d\theta \, \sigma_\ell \vartheta_\ell (1 - \vartheta_\ell) \widetilde{\varepsilon}_\ell (\widetilde{j}_\ell^E)^2 = \sum_\ell \int d\theta \, \rho_\ell (1 - \vartheta_\ell)(\widetilde{j}_\ell^E)^2. \tag{3.7}$$

Here we have used several identities which are described in Refs. [15, 16].

In Fig. 1 we show results for the thermal Drude weight $D_{\text{th}} = \langle (\mathcal{J}^E)^2 \rangle / 2NT^2$ for anisotropies $\Delta = \cos(\pi/m)$ as a function of temperature. At low temperatures one finds that both the thermal Drude weight $D_{\text{th}}(T)$ and the specific heat $C(T)$ scale linearly with temperature

$$D_{\text{th}} = \frac{\pi v}{6} T, \quad C = \frac{\pi}{3v} T, \quad \frac{D_{\text{th}}}{C} = \frac{v^2}{2}, \tag{3.8}$$

where $v = J\pi \sin \gamma / 2\gamma$ is the velocity of the elementary excitations.

We can now ask if unusual heat transport properties can be observed in experiments on spin-1/2 chain compounds in which integrability will be broken by lattice vibrations, impurities, and interchain couplings. Before discussing this point further, two important comments are in order. If one measures the thermal conductivity at finite magnetic field, then the thermal current consists of energy *and* spin current contributions: $\mathcal{J}^{\text{th}} = \mathcal{J}^E - h\mathcal{J}^s$. Experimental measurements of the heat conductivity are often done in a setup where the spin current vanishes, $\mathcal{J}^s = 0$. In this case the heat conductivity is redefined

$$K = \kappa_{\text{th}} - \frac{1}{T} \frac{(C_s^{\text{th}})^2}{\sigma_s}, \tag{3.9}$$

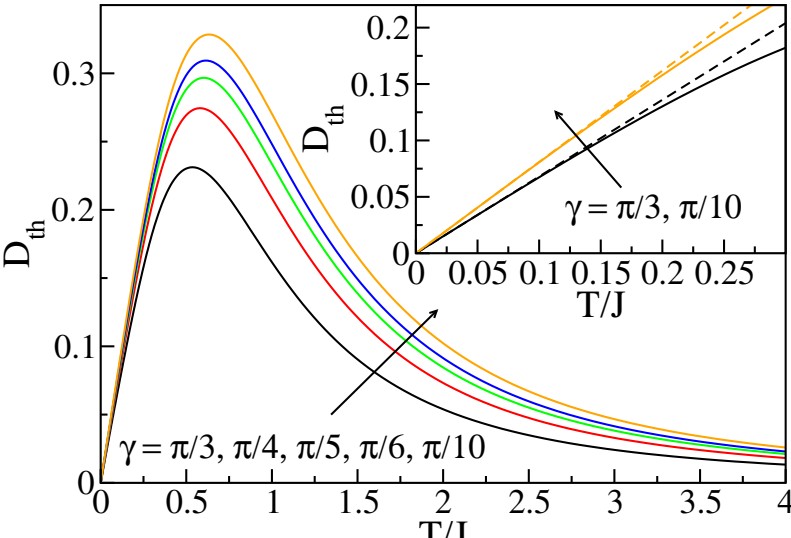

Figure 1: $D_{\text{th}}(T)$ for different anisotropies. The inset compares the full result to the low-temperature asymptotics (dashed lines), see Eq. (3.8).

where the second term is called the magnetothermal correction [17, 18]. Eq. (3.9) is based on the assumption that the relaxation times for energy and spin transport are the same which might not necessarily be true for a real material. Leaving such issues aside, one might expect that in a system which is close to an integrable one, heat currents are decaying slowly and mean free paths are long. This is indeed what seems to have been seen in a number of experiments [19–22]. For the copper-oxide spin chain compounds $Sr_2CuO_3$ and $SrCuO_2$, for example, it has been observed that the heat conductivity along the chain direction is about an order of magnitude larger than in the perpendicular directions. A natural explanation appears to be that there is heat transport due to phonons in all directions while only in the chain direction there is an additional contribution due to magnetic excitations which decays only very slowly. Obtaining a detailed understanding of the heat transport as measured experimentally is, however, a complex and still somewhat open issue. It requires an identification of the dominant relaxation processes and a formalism to incorporate such scattering mechanisms in the calculation of the thermal conductivity.

## 4 The Spin Conductivity

In this last part, I want to discuss the spin conductivity of the XXZ chain at zero magnetic field. As already mentioned, in this case none of the conserved charges (2.8) derived from the regular transfer matrix has any overlap with the spin current because of the spin-flip symmetry $\mathcal{C}$. The Mazur inequality (2.15) therefore apparently does not provide a non-zero bound. This raises the question whether or not spin transport in the integrable XXZ chain has a ballistic component. Various different approaches have been used so far to try to directly compute the Drude weight: (1) Starting from the spectral representation of the Kubo formula (2.18) and comparing this with the change of the eigenenergies $E_n$ of the Hamiltonian (1.1) when threading a static magnetic flux $\Phi$ through an XXZ ring one finds

$$D = \frac{1}{2NZ} \sum_n e^{-E_n/T} \frac{\partial^2 E_n(\Phi)}{\partial \Phi^2}\bigg|_{\Phi=0}, \tag{4.1}$$

with $Z$ being the partition function. This is a generalization of the Kohn formula [23] to finite temperatures [24]. For zero temperature, in particular, the Drude weight can be obtained simply from the ground state energy of the system with an added flux [25] leading to

$$D(T = 0) = \frac{\pi \sin \gamma}{8\gamma(\pi - \gamma)}. \tag{4.2}$$

For finite temperatures, the formula (4.1) has been used in Ref. [26] to calculate $D(T)$ for anisotropies $\gamma = \pi/m$ on the basis of the thermodynamic Bethe ansatz (TBA). The high- and low-temperature limits have then been analyzed in Ref. [27]. (2) A completely different approach is based on constructing a set of quasi-local charges—different from the ones in Eq. (2.8)—that have finite overlap with the current operator and to evaluate the r.h.s. of Eq. (2.15), see for example Refs. [2, 10–12]. A major difficulty in this approach is the evaluation of the correlators at finite temperatures. So far, only the high-temperature limit has been analyzed analytically [11] resulting in

$$\lim_{T \to \infty} 16TD = J^2 \frac{\sin^2(\pi n/m)}{\sin^2(\pi/m)} \left(1 - \frac{m}{2\pi} \sin(2\pi/m)\right). \tag{4.3}$$

Here the equal sign is only correct if the set of conserved charges used is complete which is a point which is difficult to prove. It has, however, been shown that the above result agrees with the high-temperature limit of the TBA result obtained using the Kohn formula [16] which might give us some confidence that (4.3) is not just a lower bound but indeed exhaustive. Note that the Drude weight in the high-temperature limit has a fractal character according to Eq. (4.3), while $D(T = 0)$ depends smoothly on anisotropy, see Eq. (4.2). This is opposite to our usual expectations that thermal fluctuations lead to a smoothening of the expectation values of observables as function of some parameter of the model. (3) A third approach has recently been proposed based on a generalized hydrodynamics (GHD) formulation where the continuity equations

$$\partial_t \langle Q_n \rangle + \partial_x \langle J_n \rangle = 0 \tag{4.4}$$

lead to the so-called Bethe-Boltzmann equations [28–31]

$$\partial_t \rho_{\xi,\ell}(\theta) + \partial_x \left(v_{\xi,\ell}(\theta)\rho_{\xi,\ell}(\theta)\right) = 0. \tag{4.5}$$

Here the current $J_n$ is being related to the velocity $v_{\xi,\ell}$ and density $\rho_{\xi,\ell}$ of quasi-particle excitations. $\xi = x/t$ describes a set of rays along which a local equilibration is assumed to occur. The advantage of this formulation is that also dynamics far from equilibrium can be investigated. (4) Very recently, a first principle calculation of the Drude weight starting directly from the operator expression of the spin current has been presented [16]. Here the only assumption remaining is related to the existence of a complete set of conserved charges, similar to the assumption used in the derivation of the Mazur inequality.

Since GHD has already been discussed at this Les Houches summer school, I will spend the last part of this lecture series on introducing an effective low-energy approach. In contrast to Bethe ansatz methods, this will allow to obtain a physical picture of the spin conductivity not only in integrable but also in generic spin-chain models. Furthermore, for the integrable XXZ chain we will be able to directly connect the ballistic and diffusive transport channels to each other.

## 4.1 Bosonization

Let me very briefly recapitulate the idea of bosonization. We start from the fermionic Hamiltonian (1.2) and take the continuum limit

$$c_j \to \Psi(x) = e^{ik_F x}\Psi_R(x) + e^{-ik_F x}\Psi_L(x), \quad \Psi_{R,L}(x) = \frac{1}{\sqrt{N}} \sum_{k=-\Lambda}^{\Lambda} c_{kR,L} e^{\pm ikx}, \tag{4.6}$$

where $\Psi_{R,L}$ are the right and left movers obtained by linearizing the dispersion around the Fermi points and $\Lambda$ is a momentum cutoff. The important point is that particle-hole excitations with momentum $q$ now all have the same energy, e.g., $E_R(q) = v(k+q) - vk = vq$ is independent of $k$ with $v$ being the velocity. Collective excitations of particle-hole type can therefore be represented by a bosonic operator, $\sum_k c^\dagger_{k+q} c_k \sim b_q$, and the interacting Hamiltonian (1.2), which is quartic in the fermionic operators, becomes a *quadratic bosonic theory* at low energies. The correction terms to the quadratic theory are all irrelevant in a renormalization group sense in the critical regime $-1 < \Delta < 1$. For the purpose of calculating the conductivity it is convenient to use bosonic fields which are related to the right and left movers by

$$\Psi_{R,L} \propto \frac{1}{\sqrt{2\pi\alpha}} e^{-i\sqrt{2\pi}\varphi_{R,L}}, \quad \varphi_{R,L} = \frac{1}{\sqrt{2}}(\tilde{\theta} \mp \tilde{\phi}), \tag{4.7}$$

where $\alpha \sim k_F^{-1}$ is a short-distance cutoff and we have introduced canonically conjugated fields $[\tilde{\phi}(x), \partial_{x'}\tilde{\theta}(x')] = i\delta(x - x')$. The interaction now merely leads to a rescaling of these fields, $\tilde{\phi} = \sqrt{K/2}\phi$ and $\tilde{\theta} = \sqrt{2/K}\theta$, leading to a Hamiltonian

$$H = \frac{v}{2} \int dx \left[ (\partial_x \phi)^2 + (\partial_x \theta)^2 \right] + \lambda \int dx \, \cos(\sqrt{8\pi K}\phi). \tag{4.8}$$

The first term describes the free theory while the second term with scaling dimension $2K$ represents irrelevant Umklapp scattering. The Luttinger parameter $K$ and the velocity $v$ can be determined for the integrable XXZ chain by calculating static properties such as the specific heat and the susceptibility using the field theory (4.8) and the Bethe ansatz and comparing the results. This leads to

$$v = \frac{J\pi}{2} \frac{\sqrt{1-\Delta^2}}{\arccos\Delta} = \frac{J\pi}{2} \frac{\sin\gamma}{\gamma}, \quad K = \frac{\pi}{\pi - \arccos\Delta} = \frac{\pi}{\pi - \gamma}. \tag{4.9}$$

Note that in this notation $K = 2$ at the free Fermi point $\Delta = 0$, and $K = 1$ at the isotropic point $\Delta = 1$.

The spin current density is given by $j^s = J(\Psi_L^\dagger \Psi_L - \Psi_R^\dagger \Psi_R)$ in terms of the left and right movers. Since the free bosonic Hamiltonian conserves the right and left particle densities separately, the spin current will not relax. It is thus important to also take the last term in Eq. (4.8) into account. It describes Umklapp scattering

$$\sim e^{-i2k_F(2x+1)}\Psi_R^\dagger(x)\Psi_L(x)\Psi_R^\dagger(x+1)\Psi_L(x+1) + h.c., \tag{4.10}$$

where two left movers scatter to two right movers and vice versa. In general, this term oscillates $\sim \exp(i4k_F x)$ but is non-oscillating at half-filling (zero magnetic field) where $k_F = \pi/2$. While this term is formally irrelevant for $-1 < \Delta < 1$ it can relax the current and therefore has to be treated with care.

## 4.2 Results

We now want to evaluate the Kubo formula (2.17). We can couple the fermions to the electromagnetic potential $A$ by a Peierls substitution $\Pi = \partial_x \theta \to \Pi - \sqrt{K/2\pi}A$. One then finds

$$\left.\frac{\partial H}{\partial A}\right|_{A=0} = \int dx \, j^s(x) \quad \text{with} \quad j^s = -v\sqrt{\frac{K}{2\pi}}\Pi = -\sqrt{\frac{K}{2\pi}}\partial_t \phi, \tag{4.11}$$

$$\left.\frac{\partial^2 H}{\partial A^2}\right|_{A=0} = \langle H_{\text{kin}} \rangle = \frac{vK}{2\pi}L,$$

using $\partial_x \theta = v^{-1} \partial_t \phi$. Here $L = Na$ with $a$ being the lattice constant. The second line is the diamagnetic term. The Kubo formula then reads

$$\sigma_s(q, \omega) = \frac{i}{\omega} \left[ \frac{vK}{2\pi} + \langle \mathcal{J}^s \mathcal{J}^s \rangle^{\text{ret}}(q, \omega) \right]. \tag{4.12}$$

By partial integration and using the canonical commutation relations one finds

$$\langle \partial_t \phi \, \partial_t \phi \rangle^{\text{ret}}(q, \omega) = -v + \omega^2 \langle \phi \phi \rangle^{\text{ret}}(q, \omega). \tag{4.13}$$

Putting this into (4.12) we see that the diamagnetic term is cancelled and we are left with the following simple Kubo formula for the spin conductivity within the bosonized theory

$$\sigma_s(q, \omega) = \frac{vK}{2\pi} i\omega \langle \phi \phi \rangle^{\text{ret}}(q, \omega). \tag{4.14}$$

The only quantity required to obtain the conductivity is thus the retarded correlation function of the basic bosonic field. For the free bosonic model without the Umklapp term ($\lambda = 0$ in Eq. (4.8)), we just find the standard free boson propagator

$$\langle \phi \phi \rangle^{\text{ret}}(q, \omega) = \frac{v}{\omega^2 - v^2 q^2} \tag{4.15}$$

leading to a Drude weight

$$D\delta(\omega) = \frac{1}{2\pi} \lim_{\omega \to 0} \lim_{q \to 0} \sigma'(q, \omega) = \frac{Kv}{4\pi^2} \text{Re}\left( \frac{i}{\omega + i\epsilon} \right) = \frac{Kv}{4\pi} \delta(\omega), \tag{4.16}$$

which does agree with the BA result (4.2). Note that at this level of approximation there is no regular part of the conductivity and no temperature dependence of the Drude weight. Taking into account band curvature terms will introduce a temperature dependence of the Drude weight but only the Umklapp term can lead to a relaxation of the current. For the conductivity this operator is dangerously irrelevant and will completely change the transport properties of the theory. To see this it is sufficient to calculate the propagator to second order in perturbation theory in the Umklapp scattering

$$\langle \phi \phi \rangle^{\text{ret}}(q, \omega) = \frac{v}{\omega^2 - v^2 q^2 - \Pi^{\text{ret}}(q, \omega)}, \tag{4.17}$$

where $\Pi^{\text{ret}}(q, \omega)$ is the self energy. This is a standard calculation and we just present the result here

$$\sigma(q, \omega) = \frac{vK}{2\pi} \frac{i\omega}{\omega^2 - v^2 q^2 + 2i\Gamma\omega}. \tag{4.18}$$

Here $\Gamma \sim \lambda^2 T^{4K-3}$ is a relaxation rate which vanishes for $T \to 0$. For the integrable XXZ model, $\Gamma$ can be determined exactly [32] and there are therefore no free parameters in (4.18) in this case. Here we just want to understand the physics qualitatively. Considering, in particular, the real part of the conductivity at $q = 0$ we find

$$\sigma'(\omega) = \frac{vK}{2\pi} \frac{2\Gamma}{\omega^2 + (2\Gamma)^2}. \tag{4.19}$$

The Drude weight broadens to a Lorentzian with width $\sim T^{4K-3}$ at any finite temperature. While this is in fact the expected behavior for a generic non-integrable model, we are now missing the finite-temperature Drude weight which we know does exist in the integrable XXZ chain because of the quasi-local charges which protect a part of the spin current from decaying.

This should not come as a surprise: In the derivation of the low-energy effective theory, the existence of an infinite set of (quasi-)local conserved charges $Q_n$ has not been taken into account. The requirement $[H, Q_n] = 0$ corresponds, in general, to a fine-tuning of the bosonic Hamiltonian. As has been shown in Ref. [33] this can, for example, lead to the absence of certain irrelevant terms which are kinematically allowed and therefore expected to be present in a generic model. A full understanding of the structure of the low-energy Hamiltonian for the integrable XXZ chain is, however, still lacking. Here we will instead use a different approach. If there is a conserved charge with finite overlap with the current, then we can separate this current into two parts

$$\mathcal{J}^s = \underbrace{\frac{\langle \mathcal{J}^s Q \rangle}{\langle Q^2 \rangle} Q}_{\mathcal{J}^s_{\parallel}} + \mathcal{J}^s_{\perp} . \tag{4.20}$$

Then $\mathcal{J}^s_{\perp}$ will decay due to Umklapp scattering while $\mathcal{J}^s_{\parallel}$ is protected. More formally, this approach can be implemented using a memory matrix approach, see Ref. [9,34]. The conductivity then becomes

$$\sigma'_s(\omega) = \underbrace{\frac{vK}{2} \frac{y}{1+y} \delta(\omega)}_{2\pi D_s(T)} + \underbrace{\frac{vK}{\pi} \frac{\Gamma}{\omega^2 + 4(1+y)^2 \Gamma^2}}_{\sigma'_{\text{reg}}(\omega)}, \tag{4.21}$$

with

$$\frac{y}{1+y} = \frac{\langle \mathcal{J}^s Q \rangle^2}{\langle (\mathcal{J}^s)^2 \rangle \langle Q^2 \rangle} \tag{4.22}$$

and $\langle (\mathcal{J}^s)^2 \rangle / LT = vK/2\pi$. Note that the Drude weight $D_s$ obtained from Eqs. (4.21) and (4.22) is consistent with the Mazur equation (2.15). Note, furthermore, that for $y \to \infty$ and thus $y/(1+y) \to 1$ we recover the Drude weight $D_s = vK/(4\pi)$ which therefore corresponds to the case of a fully conserved current. For $y$ finite, on the other hand, Eq. (4.21) describes a *coexistence of ballistic and diffusive transport*. Finally, we can also check that (4.21) fulfills the f-sum rule $\int d\omega \, \sigma'_s(\omega) = vK/2$.

Conversely, we can also use (4.21) to express $y$ by the Drude weight $D_s$ leading to

$$y = \frac{4\pi D_s(T)}{vK - 4\pi D_s(T)}, \quad 1 + y = \frac{vK}{vK - 4\pi D_s(T)}, \tag{4.23}$$

with $D_s(0) = vK/(4\pi)$. The regular part of the conductivity at frequency zero then reads

$$\sigma'_{\text{reg}}(\omega = 0) = \frac{vK}{4\pi} \frac{1}{(1+y)^2 \Gamma} = \frac{(vK - 4\pi D(T))^2}{4\pi vK \Gamma} . \tag{4.24}$$

For $\gamma = \pi/m$, the TBA calculations in Refs. [16, 26] have shown that at low temperatures the Drude weight behaves as $D_s(T) = D_s(0) - \alpha T^{2K-2}$ where $\alpha$ depends on the anisotropy $\gamma$. Furthermore, the relaxation rate due to Umklapp scattering can be expressed as $\Gamma = \Gamma_0 T^{4K-3}$ where $\Gamma_0$ is a function of anisotropy and is known exactly, see Ref. [9,34]. The regular part of the conductivity at low temperatures is therefore given by

$$\sigma'_{\text{reg}}(\omega = 0) = \frac{4\pi \alpha^2}{vK \Gamma_0} \frac{1}{T} . \tag{4.25}$$

We can now use the Einstein relation to define the diffusion constant

$$\mathcal{D}_s \equiv \frac{\sigma'_{\text{reg}}(\omega = 0)}{\chi_s} = \frac{8\pi^2 \alpha^2}{K^2 \Gamma_0} \frac{1}{T}, \tag{4.26}$$

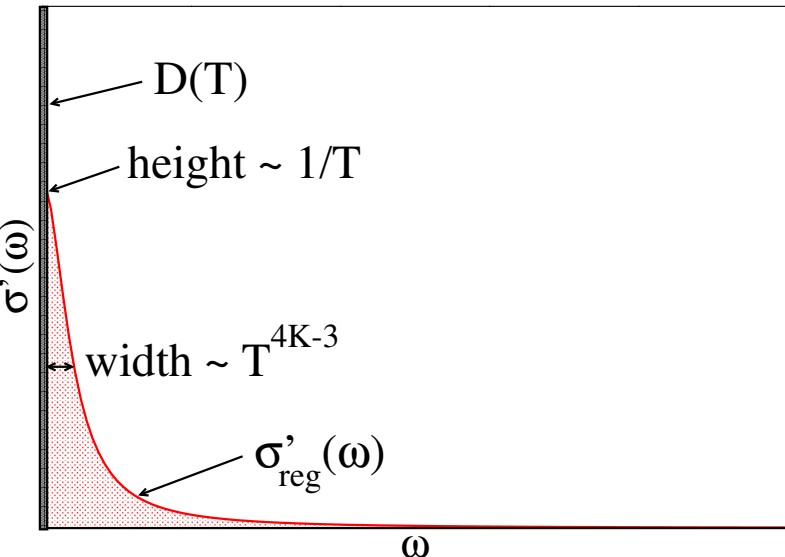

Figure 2: At finite temperatures and anisotropies $\Delta = \cos(\pi/m)$ there is a coexistence of ballistic and diffusive transport in the XXZ chain: The Drude peak sits on top of a narrow Lorentzian with width $\sim T^{4K-3}$.

where $\chi_s(T)$ is the spin susceptibility and we have used the low-temperature result $\chi_s = K/2\pi v$. The diffusion constant thus diverges as $1/T$ for $T \to 0$. Note that this derivation uses the Bethe ansatz result for anisotropies $\Delta = \cos(\pi/m)$ and is thus only valid for these discrete anisotropies. Furthermore, the relaxation rate $\Gamma = \Gamma_0 T^{4K-3}$ has only been calculated to second order in Umklapp scattering so Eq. (4.26) is only expected to be an upper bound for the exact diffusion constant at low temperatures. A formula to calculate the exact diffusion constant at anisotropies $\Delta = \cos(\pi n/m)$ has recently been conjectured in Ref. [35] based on an extension of GHD. Numerically, these predictions can be tested by calculating the diffusion constant directly from the current-current correlation function, see Eq. (2.28). In such numerical calculations, the main problem is to reach sufficiently long times to obtain reliable results for the integral over the time-dependent current-current correlation function. This problem is particularly severe at low temperatures where the current-current correlation function decays very slowly towards its long-time value $\lim_{t\to\infty}\langle \mathcal{J}^s(t)\mathcal{J}^s(0)\rangle = 2NTD_s(T)$.

Overall, we have obtained the following picture for the spin conductivity $\sigma'_s(\omega)$ of the XXZ chain at $h = 0$, anisotropy $\Delta = \cos(\pi/m)$, and small frequencies $\omega$: At $T = 0$ there is only a Drude peak $D = vK/(4\pi)$ and no regular part because Umklapp scattering is inactive. At $T > 0$, on the other hand, we have a coexistence of ballistic and diffusive transport. This coexistence manifests itself in a Drude peak on top of a narrow Lorentzian with width $\sim T^{4K-3}$ and height $1/T$. The weight of the Lorentzian is therefore $\sim T^{4K-4}$ and vanishes for $T \to 0$ if $0 < \Delta = \cos(\pi/m) < 1$. This situation is shown pictorially in Fig. 2.

# 5 Conclusion

To summarize, I have introduced the basic framework to calculate transport in the linear response regime. For integrable models, transport can be unusual in the sense that the current itself or part of the current is protected by a conservation law leading to an infinite dc conductivity even at finite temperatures. It is important to stress that the ideal conductivity in this case is not related to superconductivity: the superfluid density is zero and there is no Meisner

effect.

For the integrable XXZ spin chain in the critical regime, concrete results for the thermal and the spin conductivity at anisotropies $\gamma = \pi/m$ have been derived. These results can be easily generalized to $\gamma = n\pi/m$ with $n, m$ coprime and integer. Note that while the TBA-type approaches used here rely on having finite string lengths and can therefore not be applied if $\gamma/\pi$ is irrational, we can approximate any irrational number by a rational one to arbitrary precision. The result for the infinite temperature spin Drude weight (4.3), for example, does have a well-defined limit $16TD_s = 2\sin^2(\gamma)/3$ for $\gamma$ irrational. This suggests that the XXZ chain does show an infinite dc conductivity for all anisotropies $-1 < \Delta < 1$ and all temperatures.

Left out of these lectures has been the gapped regime of the XXZ chain, $|\Delta| > 1$, and the isotropic antiferromagnet, $\Delta = 1$. For the thermal Drude weight nothing changes qualitatively because $\mathcal{J}^E$ itself is conserved. The quasi-local charges which protect part of the spin current, on the other hand, become non-local for $|\Delta| > 1$ and the spin transport becomes diffusive [3,35]. Right at the isotropic point, $\Delta = 1$, numerical calculations point to super-diffusive transport with a dynamical critical exponent $z = 2/3$ [36]. While a mostly coherent picture of spin transport in the XXZ chain has started to emerge in the last ten years based on a number of different analytical and numerical methods, these very recent results for the isotropic point show that this picture is not quite complete yet and that this topic deserves further study.

# Acknowledgements

I am grateful to my colleagues and co-authors on a number of related research articles for sharing their insights into this subject. In particular, I would like to thank I. Affleck, R.G. Pereira, and A. Klümper. I also thank A. Urichuk for providing the data for the thermal Drude weight.

I acknowledge support by the Natural Sciences and Engineering Research Council (NSERC) through the Discovery Grants program and by the German Research Foundation (DFG) via the Research Unit FOR 2316.

# Addendum

My lecture notes above do follow closely the actual lectures given at the Les Houches summer school. As outlined in the introduction, they are not an all encompassing review of transport in integrable quantum systems but are rather meant to convey the basic concepts and to apply those concepts to a specific integrable model, the XXZ chain, for particular sets of the microscopic parameters. In this addendum, I want to provide a few additional pointers to topics and methods which were not or not extensively covered in these notes.

Apart from analytical calculations based on Bethe ansatz and low-energy effective field theories, numerical methods have also played an important role in uncovering the transport properties of quantum chains at or close to integrable points. These include, in particular, exact numerical diagonalizations [24,37], time-dependent density-matrix renormalization group calculations [9,38,39], and quantum typicality based schemes [40].

The regime of anisotropy $\Delta > 1$ is only very briefly discussed in the notes. Here I want to take the opportunity to point the reader to additional literature where this regime is discussed in much more detail [38,41,42].

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
