# Peer review of "Transport in one-dimensional integrable quantum systems"

_SciPost Physics Lecture Notes, doi:SciPost Phys. Lect. Notes 17 (2020)_

## Round 1 · Referee Report · Anonymous (Referee 1) · 2019-10-31

Report

These lectures are on a topic of actual theoretical interest, transport in integrable systems at finite temperatures, where recently significant developments are taking place.
They are also complementary to other lectures in this summer school.
The notes are well written and the presentation systematic and
well organized with enough details for a student to follow.
The key notions, thermal and spin Drude weight, Mazur inequality,
Kubo formalism are carefully introduced and the main known results criticaly discussed.
Of course the whole machinery of the TBA method is not presented, but I suppose it was discussed in other lectures.
Finally the notes are complemented with a bosonization approach
so that a complete picture of the frequency dependent
spin conductivity emerges.

I recommend publication of this manuscript as is.
  • validity: high
  • significance: high
  • originality: good
  • clarity: top
  • formatting: excellent
  • grammar: excellent

Author:  Jesko Sirker  on 2020-04-24  [id 808]

(in reply to Report 1 on 2019-10-31)

I thank the referee for their report. Basics of Bethe ansatz methods were indeed covered in other earlier lectures at the same summer school.

---

## Round 1 · Referee Report · Anonymous (Referee 2) · 2019-11-6

Strengths

Very pedagogical
Very good account of basic formalism, TBA, and field theoretical treatment

Weaknesses

Incomplete account of related literature (which, in particular, for students and newcomers to the field) is important to know about)

Without actual references, some statements about, e.g., the shortcomings of numerical
methods cannot be understood by non-experts.

The presented picture for transport is not up to date with the latest developments.

Report

The manuscript provides an introduction into the theoretical description
of transport in the spin-1/2 XXZ chains, addressing its unusual
transport properties. It focuses on the basics (i.e., general linear
response expressions, Kubo formula and the Mazur inequality) and
then covers results from Bethe-ansatz methods such as TBA and then field theory.

The text is quite well written and will certainly be of use to its target
audience (graduate students, postdocs and newcomers to the field).
Therefore, I recommend its publication in SciPost Physics
Lecture Notes, provided the following criticism is appropriately addressed.

Requested changes

1) While it is quite reasonable to restrict the presented material to specific methods and results (also considering that this is based on actual lectures), the author nevertheless needs to give a balanced account of related literature, in particular, whenever the picture presented here is not complete or conflicts with other studies.

For instance, the author is advised to account for the existence of numerical methods and their relevance by adding proper citations. A fair sampling of relevant papers should include: Znidaric, Phys. Rev. Lett. 106, 220601 (2011), Steinigeweg et al Phys. Rev. Lett. 116, 017202 (2016), Karrasch et al. Phys. Rev. Lett. 108, 227206 (2012).

2) The author correctly states that the field theory provides the generic picture in nonintegrable models, i.e., diffusion at T>0.

For integrable models, it makes a prediction that there is either diffusion or diffusion is the subleading correction next to the Drude weight (which itself cannot be obtained from field theory independently). This is summarized in Fig. 2.

The full picture, established from exact lower bounds to the diffusion constant and from generalized hydrodynamics, is more complex, though.

It is now understood that the subleading correction is superdiffusive almost everywhere in the regime Delta <1, with the exception of the so-called commensurate points.

Since the article aims at introducing the "unusual transport" in this model, these very important insights should be mentioned. Also, it is worth mentioning that so far, field theory fails to predict superdiffusion anywhere in the model (also at Delta=1), thus pointing the reader to another interesting open problem.

The relevant papers are: Ilievski et al., Phys. Rev. Lett. 121, 230602 (2018) (see footnote 67), Agrawal et al. arXiv:1909.05263

In the same context, the statement before Eq. 2.28 is not general enough: diffusion only results if the integral converges, superdiffusion, if it diverges. For completeness, subdiffusion should also be mentioned (although it is not believed to occur in clean spin chains).

3) Below Eq. (3.9): The author should also cite Louis and Gros, who first studied the term magnetothermal correction in Phys. Rev. B 67, 224410 (2003) for the spin-1/2 XXZ chain.

4) In the discussion of the experiments on page 11, one should clearly state that the notion of integrability is not needed to explain the experiments on the current level of experimental data. Moreover, it would help the reader to be directed to comprehensive reviews on the experiments, such as Physics Reports 811, 1 (2019). The largest thermal conductivities in spin chain materials were actually reported in Phys. Rev. B 81, 020405(R) (2010).

5) References for diffusion for Delta >1: The authors should also cite: Phys. Rev. Lett. 106, 220601 (2011), Phys. Rev. B 89, 075139 (2014), Phys. Rev. B 80, 184402 (2009), Phys. Rev. Lett. 107, 250602 (2011).

6) References to the current understanding of spin transport at Delta=1 should be added, in particular those also pointing to superdiffusion: Phys. Rev. Lett. 122, 127202 (2019), Phys. Rev. Lett. 122, 210602 (2019), as well as : J. Stat. Mech. 2011, P12008 (2011).

7) In particular for a text for students, it is my impression that the seminal importance of Ref. 7-9 is underrepresented in this article. These papers solved the long-standing question of whether there is a spin Drude weight at T>0 at all (in the affirmative for Delta <1) and introduced the notion of quasi-local charges, leading to (one might even say, revolutionary) new insights into the theory of Bethe ansatz integrable models. I suggest that this should be stated, where the quasi-local charges are mentioned for the first time (i.e., below Eq. 2.16).

8) The GHD result for the spin Drude weight also agrees precisely with the TBA formula and the lower bound, Eq. 4.3, see Phys. Rev. Lett. 119, 020602. This could be mentioned on page 12.

  • validity: good
  • significance: top
  • originality: good
  • clarity: high
  • formatting: perfect
  • grammar: perfect

Author:  Jesko Sirker  on 2020-04-24  [id 807]

(in reply to Report 2 on 2019-11-06)

I thank both referees for their reports.

Referee 1: Before replying to each of the points raised, let me stress again that these are Lecture Notes, i.e., they provide a written account of my lectures at the Les Houches summer school. The manuscript is not intended, and is also certainly not in practice, a full review of this field of research. I believe I have made this clear in the outline of my lecture notes. I have now also added the sentence 'Furthermore, I also note that a different approach to transport---generalized hydrodynamics---has been discussed in a separate series of lectures and will not be covered here.' in the outline. These lecture notes will be published together so it makes little sense in my view to now include a discussion of a topic with many further references which was not at all discussed in my lectures at Les Houches but was rather covered in depth in a separate series of lectures at the same summer school.

1) For the first part, see above. I have not added any references to numerical papers because i) I never discussed numerical methods at all during the lectures so giving references for a topic not covered makes little sense in my view, and ii) I believe that if I would try to add some discussion about numerical results then a fair sampling would then require to list many more paper than the three mentioned by the referee.

2) The article Phys. Rev. Lett. 121, 230602 (2018) is already cited and discussed in the conclusions. Furthermore, it is mentioned several times that only anisotropies Delta=cos(pi/m) are considered. I now refer to the results away from the commensurate points a few more times throughout the notes, see list of changes.

Below Eq. (2.28), in particular, I added a sentence about superdiffusion and the divergence of the integral.

3) The reference has been added.

4) The reference Phys. Rev. B 81, 020405(R) (2010) has been added. I note that the authors do interpret their results as clear indication of ballistic transport in the underlying spin-1/2 Heisenberg model. We know that many properties of these systems are very well described by the spin-1/2 Heisenberg model. It is therefore also a natural starting point to discuss the transport properties. I have made it clear, however, that a detailed understanding is still lacking.

5) The case Delta>1 is not part of the lectures. It is briefly discussed in the conclusions and some references are given.

6) Again, this case was not part of the lectures. It is briefly discussed in the conclusions and some references are given.

7) The results in [7-9] do confirm the results derived by Zotos in 1999. They have been without a doubt very important for our understanding of transport in the XXZ chain but I would not subscribe to the referee's point that they have solved the question whether or not there is a spin Drude weight at T>0. They are partly independent from the approach used by Zotos while both share some other possible issues. So either both approaches are correct in which case the spin Drude weight for T>0 was already established by Zotos or they are both incorrect (which seems much less likely) in which case the problem would still be open. In any case, the references are cited and the results discussed.

8) GHD is discussed in a separate series of lectures in the same volume.

List of changes:

1) Outline: I added the sentence 'Furthermore, I also note that a different approach to transport---generalized hydrodynamics---has been discussed in a separate series of lectures and will not be covered here.'

2) End of chapter 1: ' and discuss the general picture emerging from these calculations as well as their limitations.'

3) End of introduction to chapter 2: 'While we concentrate on these specific anisotropies in the following, we note that the results can be generalized to all commensurate anisotropies $\gamma = n\pi/m$ with $n,m$ coprime. On the other hand, it has been argued that ballistic transport possibly coexists with superdiffusion at incommensurate anisotropies while transport is entirely superdiffusive at the isotropic point, $\Delta=1$ [34].

4) Below Eq. (2.28): 'We note that we have assumed here that the integral in Eq.~(2.28) is convergent. If this is not the case, then the additional channel is superdiffusive. As indicated before, this possibly happens at incommensurate anisotropies but will not be discussed here further.'

5) Below Eq. (2.28): 'It provides a strict lower bound---possibly even exhaustive---for rational $\gamma/\pi$ and thus proof that ballistic transport for these anisotropies indeed persists at finite temperatures.'

6) Reference to Louis, Gros below Eq. (3.9) added.

7) Experimental reference add below Eq. (3.9)

---

## Round 4 · Referee Report · Anonymous (Referee 2) · 2020-5-8

Strengths

See 1st report

Weaknesses

See 1st report

Report

In the revised version, the author has implemented some but not all of the suggestions and
requests from the previous report.

One should stress that the points from the previous report could have been implemented
without much effort, by adding or modifying a few sentences and adding references.

The author dismisses the suggestions using these arguments, paraphrased in my own words:

(i) in the author’s view, the notes should only describe the actual contents
of the author's lecture
(ii) some aspects are mentioned in other lectures of the same school (e.g., GHD) and hence do
not require mentioning
(iii) a fair sampling of e.g. numerics would require "to list many more paper than the three
mentioned by the referee."
(iv) certain topics/methods were not discussed in the lectures.

These arguments are not convincing. Point (i): we are discussing a manuscript that is accessible individually in SciPost and on the arXiv and that can and should be viewed as a stand-alone
article. There is no reason why a lecture and a manuscript should be their exact mirror: one is
accessible to a limited audience, the manuscript is an article available to the public and if
peer-reviewed, subject to the usual standards.

Point (ii): While GHD is covered in other parts of the school, other aspects (e.g., contributions
from numerics) were maybe not covered there. We are not debating adding extensive
discussions, but only references that allow the reader to explore topics on their own that are not covered in the lecture notes.

(iii) If it takes a long list of reference to provide a fair sampling of contributions from numerical methods, then so be it. My comment from the first report did not imply exclusiveness of the
suggested references.

(iv) Precisely because the contents of the lectures cannot cover the whole field, it is essential
that references to those parts that were not covered are provided in the text, in particular, for
students and researchers from outside of the field.

Perhaps the guidelines of the Physical Review on Referencing might be helpful here
(see https://journals.aps.org/prb/authors/editorial-policies-practices):

"Manuscripts must provide proper citations to pertinent earlier work and credit significant contributions by non-authors. Readers benefit from complete referencing, which correctly contextualizes the work in regards to related research. Authors should make every effort to ensure that their citations to previously published work are comprehensive at the time of submission. These citations can include references to books and references to published conference proceedings that contain more than abstracts. Prior to publication, authors should add citations to works published during the course of the review process."

In my understanding, these are generally and widely accepted criteria and are of particular
significance for a review or introductory article such as this manuscript.

The author is therefore advised to implement the remaining suggestions 1), 4), 5), 6), 7) and 8) from the first report – see below for further details.

Furthermore, the author is encouraged the to re-examine the whole manuscript in view of the
guidelines quoted above.

Requested changes

More detailed comments on the points from the first report that were not implemented:

Ad 1) Numerical methods are actually mentioned and discussed in the manuscript, see the bottom of page 14:

"Numerically, these predictions can be tested by calculating the diffusion constant directly from the current-current correlation function, see Eq. (2.28). In such numerical calculations, the
main problem is to reach sufficiently long times to obtain reliable results for the integral over
the time-dependent current-current correlation function."

Without proper references, this text is useless to readers not familiar with the field or the
literature. The author must provide references to pertinent literature.

In this context, see also page 4: where Ref 34 is cited: Super-diffusion at the isotropic point was first suggested by Znidaric, Phys. Rev. Lett. 106, 220601 (2011).

Ad 4) Discussion of experiments, pages 9/10: Why not cite the most recent review by Hess here? See the first report.

Also, on page 9: "Obtaining a detailed understanding of the heat transport as measured experimentally is, however, a complex and still somewhat open issue. It requires an identification of the dominant relaxation processes and a formalism to incorporate such scattering mechanisms in the calculation of the thermal conductivity."

There is pertinent theoretical literature here as well that should be cited, by Rosch, Chernyshev, Rozhkov, and others. The statement is a dead end for the reader if left without references.

Ad 5) Page 14: The author should add the references mentioned in the first report about Delta >1, Otherwise, the history and contributions of a number of researchers are not properly accounted for. See the sentence

"The quasi-local charges which protect part of the spin current, on the other hand, become non-local for Delta > 1 and the spin transport becomes diffusive"

Ad 6) See the previous report.

Ad 7) The key (and seminal) contribution of the work of Prosen et al. Ref. 7-9 was to discover the quasi-local charges and to relate them to the Drude weight. This contribution should be acknowledged.

For instance, see page 6: Page 6: "These charges are sometimes called quasi-local and play an important role in understanding the spin transport properties of the XXZ chain."

Here, the author should state that these charges were discovered in Prosen PRL 2011 Prosen and Ilievski PRL 2013 (Refs. 7, 8).

Ad 8) See the previous report.

  • validity: high
  • significance: high
  • originality: good
  • clarity: high
  • formatting: perfect
  • grammar: perfect

Author:  Jesko Sirker  on 2020-05-31  [id 843]

(in reply to Report 1 on 2020-05-08)

In the second report, the referee repeats the points from the first
report. While it is clear that the referee is very passionate about
the requested changes, the report, in my view, does cross a line which
is not helpful. Besides, the report misses that many of the requested
changes have actually been implemented. For the remaining points where
we do disagree, it would have been appropriate, in my view, to reflect
on my arguments instead of dismissing them outright. It is possible to
disagree in a respectful manner.

Since I have already replied to all the points raised in my first
reply and have nothing to add, I will instead summarize some thoughts
in order to move forward. One the one hand, I have some general
remarks about lecture notes which are perhaps worth considering for
the future. On the other hand, I suggest a compromise for this
particular case which, hopefully, allows to bring this long process to
a conclusion.

Let me start with my general remarks: The Les Houches Lecture Notes
are a well established format which is different from a review
journal. Being pedagogical and also reflecting the personal views of a
researcher of a particular research field are, in my view, what makes
them valuable and is why they are often recommended to graduate
students. For the graduate students which attended the school it is,
furthermore, important that the notes do reflect what was actually
taught. For this reason it should also be clear that lecture notes can
never be fully up-to-date. While this school (held in 2018) might be
an extreme case, there will likely always be some delay between the
actual lectures and the date when the corresponding notes are
published. I do not think that there should be any pressure on the
lecturers to include material published after the school was
concluded. I personally feel that it would be a shame to move away
from the established, successful format and to effectively turn the
Les Houches Lecture Notes into yet another review journal.

Coming back to the manuscript at hand: I am facing a bit of a
conundrum here because I am convinced that my lecture notes are
completely in line with the established format. A check of a sampling
notes from previous schools on related topics confirms this. In
particular, I note that most full lecture courses (mine was only a
half a course) - resulting in lecture notes of 30-80 pages - list
40-50 references (of course, there are some exceptions with much fewer
or many more as well). In the latest version of my notes, 41
references are included which therefore is already near the upper end
of the spectrum.

In the main text, I do not want to move away from presenting the
material in a pedagogical way and from selecting the references
accordingly. I also believe that it would be counterproductive to
extend the notes to topics which were not covered in the actual
lectures. As a compromise and in order to move forward, I have now
added an addendum at the end of the notes where additional references
for topics not or only very briefly discussed in my lectures are
given. This includes, in particular, numerical studies of transport in
integrable models and studies of the regime with anisotropy Delta > 1.

I hope that the manuscript in its current form can now be accepted and
published without further delay.

---

## Round 4 · List of Changes

1) Outline: I added the sentence 'Furthermore, I also note that a different approach to transport---generalized hydrodynamics---has been discussed in a separate series of lectures and will not be covered here.'

2) End of chapter 1: ' and discuss the general picture emerging from these calculations as well as their limitations.'

3) End of introduction to chapter 2: 'While we concentrate on these specific anisotropies in the following, we note that the results can be generalized to all commensurate anisotropies $\gamma = n\pi/m$ with $n,m$ coprime. On the other hand, it has been argued that ballistic transport possibly coexists with superdiffusion at incommensurate anisotropies while transport is entirely superdiffusive at the isotropic point, $\Delta=1$ [34].

4) Below Eq. (2.28): 'We note that we have assumed here that the integral in Eq.~(2.28) is convergent. If this is not the case, then the additional channel is superdiffusive. As indicated before, this possibly happens at incommensurate anisotropies but will not be discussed here further.'

5) Below Eq. (2.28): 'It provides a strict lower bound---possibly even exhaustive---for rational $\gamma/\pi$ and thus proof that ballistic transport for these anisotropies indeed persists at finite temperatures.'

6) Reference to Louis, Gros below Eq. (3.9) added.

7) Experimental reference add below Eq. (3.9).

---

## Round 5 · List of Changes

Addendum added, where numerical methods and the Delta > 1 regime are
briefly discussed. The addendum includes pointers to a sampling of
literature on these topics; Refs. [36-41] have been added.
briefly discussed. The addendum includes pointers to a sampling of
literature on these topics; Refs. [36-41] have been added.

---

## Editorial Decision

published